# Impacts of Fengyun-4A and Ground-Based Observation Data Assimilation on the Forecast of Kaifeng's Heavy Rainfall (2022) and Mechanism Analysis of the Event

Jianbin Zhang [1], Zhiqiu Gao [1,2,*], Yubin Li [1] and Yuncong Jiang [1]

1 School of Atmospheric Physics, Nanjing University of Information Science & Technology, Nanjing 210044, China; 20211103015@nuist.edu.cn (J.Z.); liyubin@nuist.edu.cn (Y.L.); 20211103004@nuist.edu.cn (Y.J.)
2 State Key Laboratory of Boundary Layer Physics and Atmospheric Chemistry, Institute of Atmospheric Physics, Chinese Academy of Sciences, Beijing 100081, China
* Correspondence: zgao@nuist.edu.cn

**Abstract:** The advancement of Numerical Weather Prediction (NWP) is pivotal for enhancing high-impact weather forecasting and warning systems. However, due to the high spatial and temporal inhomogeneity, the moisture field is difficult to describe by initial conditions in NWP models, which is the essential thermodynamic variable in the simulation of various physical processes. Data Assimilation techniques are central to addressing these challenges, integrating observational data with background fields to refine initial conditions and improve forecasting accuracy. This study evaluates the effectiveness of integrating observations from the Fengyun-4A (FY-4A) Advanced Geosynchronous Radiation Imager (AGRI) and ground-based microwave radiometer (MWR) in forecasts and mechanism analysis of a heavy rainfall event in the Kaifeng region of central China. Our findings reveal that jointly assimilating AGRI radiance and MWR data significantly enhances the model's humidity profile accuracy across all atmospheric layers, resulting in improved heavy rainfall predictions. Analysis of the moisture sources indicates that the storm's water vapor predominantly originates from westward air movement ahead of a high-altitude trough, with sustained channeling towards the rainfall zone, ensuring a continuous supply of moisture. The storm's development is further facilitated by a series of atmospheric processes, including the interplay of high and low-level vorticity and divergence, vertical updrafts, the formation of a low-level jet, and the generation of unstable atmospheric energy. Additionally, this study examines the influence of Tai-hang Mountain's terrain on precipitation patterns in the Kaifeng area. Our experiments, comparing a control setup (CTL) with varied terrain heights, demonstrate that reducing terrain height by 50–60% significantly decreases precipitation coverage and intensity. In contrast, increasing terrain height enhances precipitation, although this effect plateaus when the elevation increase exceeds 100%, closely mirroring the precipitation changes observed with a 75% terrain height increment.

**Keywords:** assimilation; FY-4A AGRI; ground-based microwave radiometer; heavy rainfall

## 1. Introduction

The Intergovernmental Panel on Climate Change (IPCC) Sixth Assessment Report highlights a global trend towards increased frequency and intensity of extreme precipitation events due to global warming [1]. Over the past five decades, China has experienced a noticeable rise in both the duration and volume of extreme precipitation, underscoring the urgent need for in-depth research into the characteristics and causes of these events [2]. The significance of such research is amplified by the profound impact that heavy rainfall has on economic stability and public welfare, leading to extensive domestic and international studies [3–10]. China, situated in the East Asian monsoon region, presents a unique case for the study of heavy rainfall due to its complex topography and the distinct distribution of climate patterns over time and space. The variability in heavy rainfall events across different

regions is closely linked to the summer monsoon belt's positioning, highlighting the importance of regional analyses. Notable contributions to this field include Gao et al. [11], who reviewed advances in understanding the mechanisms, numerical simulations, and prediction methods of heavy rainfall and Bao [12], who analyzed the large-scale circulation patterns associated with persistent heavy rainfall in China, identifying common and distinct features of these events. Henan Province, a key agricultural area prone to meteorological disasters, experienced two significant rainfall events: the "758" Zhumadian event of 1975, with record daily precipitation, and the "720" Zhengzhou event of 2021, which set new records for hourly and daily precipitation rates and resulted in substantial human and economic losses. The 1975 event was primarily attributed to the interaction between a typhoon and a westerly trough [13], while the 2021 event was analyzed by Gao et al. [10], who pointed out the critical role of the abnormal northward extension of the subtropical high-pressure system and a significant water vapor transport by the "Cempaka" typhoon, compounded by geographic features like the Funiu and Taihang Mountains.

The advancement of Numerical Weather Prediction (NWP) is pivotal for enhancing high-impact weather forecasting and warning systems. Recent progress in this field has been driven by advancements in numerical simulation theory, increased computational power, and expanded observational capabilities [14]. Despite these developments, the forecasting of severe mesoscale rainstorms remains challenging due to the nonlinear nature of atmospheric processes, uncertainties in initial conditions, and complexities in physical process parameterization [15–17]. Data assimilation (DA) techniques are central to addressing these challenges, integrating observational data with background fields to refine initial conditions and improve forecasting accuracy [18–20].

The ongoing evolution of DA methods, alongside the assimilation of data from diverse sources, underscores the importance of leveraging high spatio-temporal resolution observations, such as those provided by geostationary satellites, to enhance NWP models [17,21–26]. The launch of Fengyun-4A (FY-4A) in 2016 marked a significant milestone in China's meteorological observation capabilities. FY-4A, equipped with advanced instruments like the Geostationary Interferometric Infrared Sounder (GIIRS) and the Lightning Mapping Imager (LMI), offers unprecedented atmospheric data quality, including improved spectral bands and spatial resolutions compared to its predecessors [27,28]. This has enabled more accurate atmospheric analyses, particularly in the assimilation of AGRI infrared radiance and ground-based microwave radiometer (MWR) data, as demonstrated by studies on the prediction of short-duration heavy rainfall events [29–31]. Their findings indicate that the joint assimilation of these data sources significantly enhances moisture distribution accuracy across atmospheric layers, thereby improving the precision of intense precipitation forecasting. However, challenges remain, particularly in the assimilation of all-sky radiance data, which is complicated by the sensitivity of infrared radiances to cloud processes and the resultant impact on forecast predictability [32–35].

A microwave radiometer (MWR) is a passive ground-based detection device that provides uninterrupted autonomous operation and real-time accurate atmospheric monitoring in all weather conditions [36,37]. Additionally, the continuous profiling of temperature and humidity data obtained from the MWR effectively supplements sounding observations. Consequently, integrating MWR data into Numerical Weather Prediction (NWP) models can enhance weather forecasts. An illustration of this is the utilization of 3-Dimensional Variational Assimilation (3DVAR) with data collected from seven ground-based MWRs during a heavy rainfall event in Beijing [38]. Nonetheless, the current utilization of MWR observations, especially in numerical models, remains inadequate.

Due to the high spatiotemporal variability, accurately describing the moisture field poses a challenge in NWP models [39]. Nevertheless, moisture stands out as a fundamental thermodynamic parameter crucial for simulating diverse physical phenomena. Within cloud microphysics, radiative transfer, and cumulus convection processes, the tri-state phase change of water linked with moisture can significantly impact the atmospheric dynamic and thermodynamic settings. Moreover, atmospheric moisture plays a pivotal

role in initiating and fostering deep convection. Hence, errors in initial moisture can directly impact cloud distribution simulations. AGRI channels 9–10 serve as water vapor absorption tools that reflect the real moisture status in the mid-to-upper troposphere [40]. While AGRI falls short in monitoring planetary boundary layers (PBL), this limitation can be compensated by employing ground-based MWR tailored for profiling PBL observations. By jointly assimilating data from FY-4A AGRI and ground-based MWR, it becomes possible to rectify the initial moisture conditions in model simulations, a critical factor for enhancing weather forecast precision [41]. Kaifeng, located in the eastern plain of Henan Province of Central China (Figure 1a,b), with its low-lying terrain and vulnerability to urban flooding, requires accurate weather prediction models to mitigate risks associated with heavy rainfall events. Previous studies on heavy rainfall in Kaifeng mainly focused on the analysis of meteorological conditions and physical quantity diagnostics related to rainfall occurrence [42–44]. There is an urgent need to utilize numerical simulation methods to investigate the mesoscale system structure and formation mechanisms during the occurrence and maintenance phase of heavy rainfall in this region.

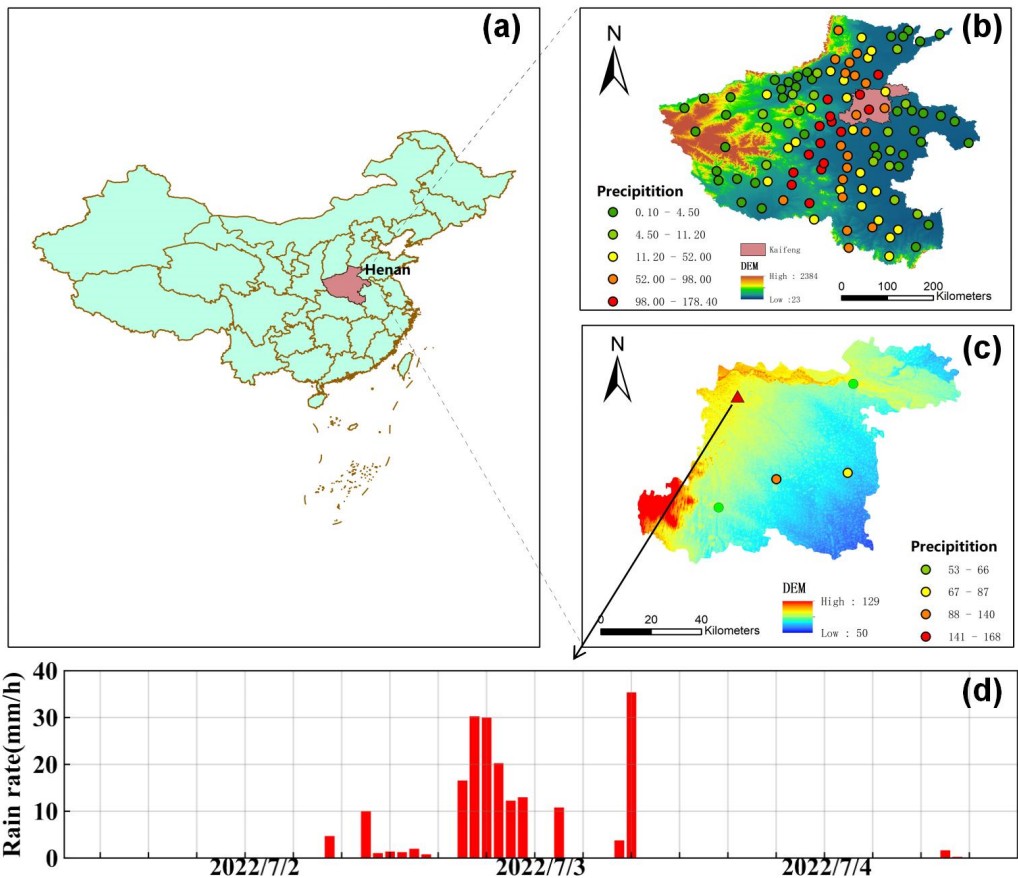

**Figure 1.** Location of Henan region (**a**); distribution of 24 h accumulated precipitation at national stations in Henan and Kaifeng (**b**,**c**); time series of hourly rainfall recorded at Longting station (Station No. 57091) in Kaifeng from 00:00 to 24:00 on 2–4 July 2022 (**d**).

The motivation of this study is to build on the work of Shi et al. [29] and employ the WRFDA v4.3 model with the AGRI observation operator and concurrent assimilation of MWR data, enhancing the initial and simulated moisture conditions for more precise convective rainfall forecasts of Kaifeng's heavy rainfall. By examining the structural characteristics and influencing mechanisms of heavy rainfall from various perspectives—including water vapor, dynamics, and topography—this research not only advances our understanding of extreme precipitation events under complex terrain but also provides a solid foundation for future forecasting and warning efforts.

## 2. Data and Methods

### 2.1. The Heavy Rainfall Case

On 3 July 2022, Henan Province experienced a significant storm weather event characterized by intense precipitation, particularly in central Henan. Analysis of the 24 h cumulative precipitation distribution highlighted a pronounced heavy rainfall belt stretching across the Nanyang–Pingdingshan–Zhengzhou–Kaifeng–Xinxiang (NY-PDS-ZZ-KF-XX) line, displaying a northeast to southwest orientation (Figure 1b). Within this belt, Kaifeng stood out with two major rainfall centers located in Tongxu County (TX) and Lankao County (LK), each recording precipitation levels exceeding 140 mm. In contrast, areas such as Qixian (QX), Weishi (WS), and Longting District (LT) experienced comparatively lower rainfall intensities (Figure 2). This event showcased a stark spatial disparity in precipitation across eastern and mid-western Kaifeng, emphasizing the event's strong localized and extreme nature.

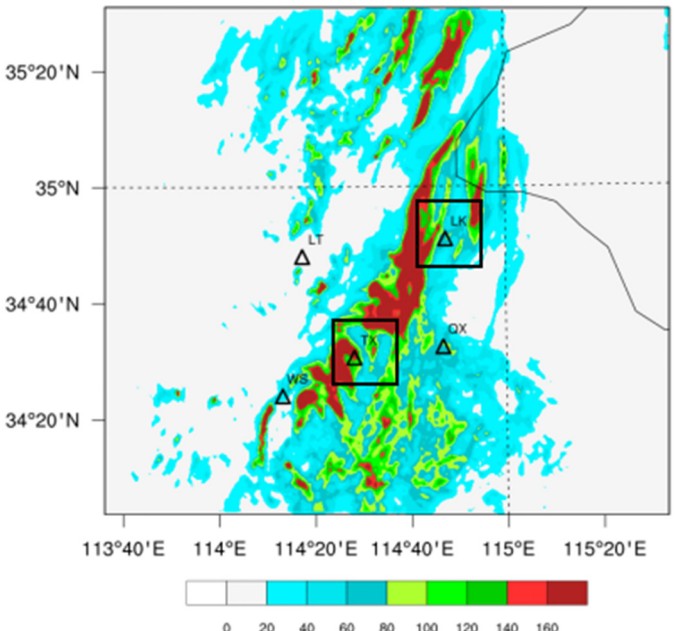

**Figure 2.** Distribution of 24 h accumulated rainfall starting from 00 UTC on 3 July 2022 in Kaifeng area.

The most intense phase of rainfall occurred between 08:00 on July 3rd and 00:00 on July 4th (Figure 1d). Unlike previous significant rainfall events in the region, such as the "758" and "720" events, this particular downpour did not benefit from the substantial water vapor contributions typically provided by typhoons. Despite meteorological predictions identifying the likely affected areas and general location of the heavy rain, the forecasts significantly underestimated the actual intensity and extent of the precipitation. This discrepancy led to substantial impacts on local travel and safety, highlighting the critical need for a deeper investigation into the mechanisms driving this heavy rain event and the factors contributing to the forecasting challenges encountered.

### 2.2. The Model

WRFDA, the assimilation system used in the Weather Research and Forecasting (WRF) model, is widely used in both scientific research and operational work. It enables the incorporation of diverse observational data, including surface weather stations, radiosondes, radars, and satellites. WRFDA offers various assimilation methods, such as 3D Variational (3DVAR), 4D Variational (FRI-4DVAR), Multi-Resolution 3rd/4th-Dimensional Variational (MRI-3/4DVAR), Ensemble Extended Kalman Filter (ETKF), and Mixed Data Assimilation

Hybrid-3D/4DEnVar. For this study, we selected the 3DVAR method for assimilation experiments and research. The equation is presented below [29]:

$$J(x) = \frac{1}{2}(x - x_b)^T B^{-1}(x - x_b) + \frac{1}{2}(y - H(x))^T R^{-1}(y - H(x))$$ (1)

In Equation (1), $x$ is the atmospheric state vector, $x_b$ is the background state vector, $H$ is the observation operator, **y** is the observation vector, and $B$ and $R$ are the covariance matrices for background error and observation error, respectively. The background error covariance matrix defined in Equation (2) is:

$$B = \overline{(x_b - x_t)(x_b - x_t)^T}$$ (2)

where $x_t$ is the true state vector, and the overbar represents an average of a number of forecasts.

By definition, acquiring precise values for $R$ and $B$ necessitates having complete knowledge of the genuine atmospheric conditions across the entire model's computational grid at all times. However, achieving such knowledge is impracticable, and as a result, both matrices must be estimated. In practice, it is common to assume that the $R$ matrix is diagonal, i.e., the observation errors are assumed to be uncorrelated and possess variances that are prescribed empirically.

RTTOV (radiative transfer for TOVS) v12.1 is a fast radiative transfer mode software package developed by NWP/SAF (Numerical Weather Prediction/Satellite Application Facilities) under EUMETSAT (European Organisation for the Exploitation of Meteorological Satellites), which is widely used in various numerical assimilation modules, primarily for the assimilation of satellite data from the visible, infrared, and microwave channels of passive detection satellites. It provides valuable information on atmospheric states. Therefore, we chose to utilize RTTOV v12.1 as the observation operator for FY-4A radiance data [29].

### 2.3. AGRI Radiance and MWR Data

The new-generation geostationary satellite FY-4A from China is equipped with an Advanced Geostationary Radiance Imager (AGRI). The AGRI includes 14 channels in the visible, near-infrared, and infrared (IR) spectral bands. The spectral coverage, spectral bands, spatial resolution, and main applications for each channel are given in Table 1 [28,29]:

**Table 1.** Specifications for AGRI on board FY-4A.

| Spectral Coverage | Channel Number | Spectral Band (µm) | Spatial Resolution (km) | Main Application |
|---|---|---|---|---|
| VIS/NIR | 1 | 0.45–0.49 | 1 | Aerosol, visibility |
| | 2 | 0.55–0.75 | 0.5 | Fog, clouds |
| | 3 | 0.75–0.90 | 1 | Aerosol, vegetation |
| | 4 | 1.36–1.39 | 2 | Cirrus |
| | 5 | 1.58–1.64 | 2 | Cloud, snow |
| | 6 | 2.10–2.35 | 2 | Cloud phase, aerosol, vegetation |
| | 7 | 3.50–4.00 | 2 | Clouds, fire, moisture, snow |
| | 8 | 3.50–4.00 | 4 | Land surface |
| Midwave IR | 9 | 5.8–6.7 | 4 | Upper-level water vapor |
| | 10 | 6.9–7.3 | 4 | Midlevel water vapor |
| Longwave IR | 11 | 8.0–9.0 | 4 | Volcanic ash, cloud-top phase |
| | 12 | 10.3–11.3 | 4 | SST, LST |
| | 13 | 11.5–12.5 | 4 | Clouds, low-level WV |
| | 14 | 13.2–13.8 | 4 | Clouds, air temperature |

Compared to the Visible Infrared Spin-Scan Radiometer (VISSR) onboard the FY-2 geostationary satellite, the Advanced Geostationary Radiation Imager (AGRI) on the FY-4A satellite boasts enhanced capabilities, including a greater number of spectral bands and higher spatial and temporal resolutions. These improvements enable AGRI to deliver more accurate information about the atmospheric state. Despite its advanced features, there has

been a notable absence of literature documenting the direct assimilation of FY-4A AGRI data. However, some researchers have discussed their work on assimilating radiance data from the AGRI water vapor (WV) channels within the Global/Regional Assimilation and Prediction System (GRAPES) model. Their preliminary results suggest that incorporating the clear-sky WV channel data from AGRI has a positive, albeit modest, effect on improving the moisture field accuracy in the mid-troposphere over East Asia [28].

AGRI stands out for its high temporal resolution, which makes it capable of conducting a complete disk observation in approximately 15 min. This facilitates the generation of one full-disk image every hour, with three consecutive images produced every 3 h, resulting in a total of 40 full-disk images each day, including detailed imagery of the Chinese region [45].

Figure 3 showcases the Jacobian functions related to temperature for AGRI's infrared (IR) channels, based on the standard atmosphere profiles by RTTOV (excluding Channels 1–7 due to their limited simulation capacity for visible and near-infrared bands). These Jacobian functions illustrate how the top-of-atmosphere (TOA) radiance changes in response to variations in atmospheric or surface conditions. Notably, channels 9 and 10 (ch9 and ch10) of AGRI are sensitive to temperatures within the 400–600 hPa layer, acting as WV absorption channels. The observed brightness temperature (BT) and the model-simulated BT for these channels show only minor discrepancies, indicating their reliability in capturing moisture profiles essential for accurate severe rainstorm simulations. Therefore, our experiments focus on assimilating data from ch9 and ch10, utilizing full-disk level 1 (L1) raw data with a spatial resolution of 4 km, leveraging their potential to enhance weather prediction models.

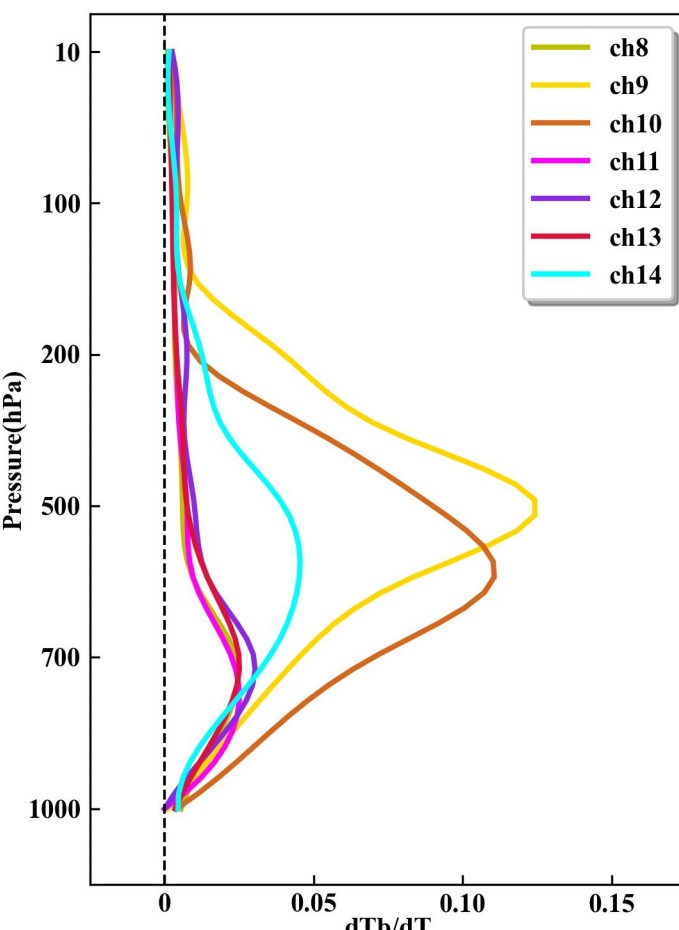

**Figure 3.** Jacobian functions of temperature of ch8-ch14 AGRI IR channels calculated by RTTOV.

The microwave radiometer (MWR) is instrumental in measuring atmospheric emissions within the microwave spectrum, deriving crucial parameters such as temperature,

relative humidity, and absolute humidity. There has been considerable effort in developing retrieval techniques for ground-based MWRs, which, when applied to Numerical Weather Prediction (NWP), have shown accuracy levels comparable to those of radiosonde measurements [46]. Research incorporating MWR-derived profiles has underscored their utility in improving local weather forecasts [38,47–50]. The RPG-HATPRO MWR system features 14 channels: 7 K-band channels dedicated to water vapor profile retrieval and 7 V-band channels (within the oxygen band) for temperature profile retrieval. The inversion software v1.0 supplied by the manufacturers is employed to generate level 2 products, including detailed temperature and relative humidity profiles [50,51].

*2.4. Quality Control*

Within the framework of the 3DVar data assimilation method, it is essential to assume unbiased Gaussian distributions for both observation and background errors [52]. Determination of the atmospheric state relies on an optimal, unbiased linear method dependent on the background conditions and their error covariance matrix alongside the observation data [53,54]. As such, data quality control (QC) is paramount in fulfilling the assimilation system's requirements, with the accuracy of numerical predictions being directly influenced by the QC process [55,56].

Infrared (IR) radiance from satellites is limited to data capture above cloud tops, as it cannot penetrate clouds. Additionally, rapid radiative transfer models often struggle with accurately simulating brightness temperature (BT) in IR channels within cloudy regions. Therefore, this study excluded cloud-affected pixels from AGRI's level 1 radiance data, assimilating only clear-sky radiance. Cloud detection leveraged the National Satellite Meteorological Center's Cloud Mask product, with a 4 km resolution at level 2, categorized into confidently clear, probably clear, probably cloudy, and confidently cloudy tiers. Only "confidently clear" pixels were assimilated into this study [57,58].

To avoid severe pixel distortions, only satellite zenith angles under 60° were selected. The high-resolution AGRI radiance data pose challenges such as increased computational costs and elevated observation error correlation. A thinning mesh of 20 km was employed to mitigate these issues, aggregating observations within each grid cell to approximate 3–6 times the horizontal resolution of the WRF model used in this study, thus minimizing model noise [57–59].

Furthermore, additional QC measures were applied during the AGRI assimilation process to address systematic errors, including the exclusion of channels with mixed surface types, rejection of observation pixels where satellite zenith angles exceed 60°, dismissal of observations if the bias-corrected innovation exceeds three times the observation error under clear-sky conditions or 15 K, and the use of the AGRI L2-level Cloud Mask product for cloud detection. This rigorous QC approach significantly improved the AGRI data quality, focusing on assimilating only the non-cloudy pixels [29].

$$\widetilde{H}(x,\beta) = H(x) + \boldsymbol{\beta_0} + \sum_{i=1}^{N_p} \beta_i p_i \tag{3}$$

In Equation (3), $H$ and $\widetilde{H}(x,\beta)$ represent the observation operators before and after the bias correction, respectively; $x$ is the atmospheric state vector; $\boldsymbol{\beta_0}$ signifies the constant bias term; $p_i$ and $\beta_i$ are the i-th predictor and the corresponding bias correction coefficients, respectively; and $\beta_i$ is automatically updated as the assimilation iteration progresses. The bias correction coefficients are usually assumed to be channel-dependent and can be estimated offline or updated adaptively within a variational minimization process by including them in the state vector. The latter is the variational bias correction (VarBC). WRFDA's VarBC implementation includes seven predictors: the scan position, the square and cube of the scan position, 1000–1300 and 200–250 hPa layer thicknesses, surface skin temperature, and total column water vapor. For AGRI radiance assimilation in this study, the scan position can be the satellite zenith angle of the pixel. The training of the bias

correction consists of finding the vector $\boldsymbol{\beta}$ that allows the best fit between the NWP fields $\boldsymbol{x}$ and the observations. This is obtained by minimizing the following cost function:

$$J(\boldsymbol{\beta}) = \frac{1}{2}[\boldsymbol{y} - \widetilde{H}(\boldsymbol{x}, \boldsymbol{\beta})]^T[\boldsymbol{y} - \widetilde{H}(\boldsymbol{x}, \boldsymbol{\beta})] \tag{4}$$

The cost function, to be minimized with respect to the bias parameters $\boldsymbol{\beta}$ and the model state $\boldsymbol{x}$, is:

$$\begin{aligned} J(\boldsymbol{x}, \boldsymbol{\beta}) = \tfrac{1}{2}[\boldsymbol{y} - \widetilde{H}(\boldsymbol{x}, \boldsymbol{\beta})]^T \boldsymbol{R}^{-1}[\boldsymbol{y} - \widetilde{H}(\boldsymbol{x}, \boldsymbol{\beta})] + \tfrac{1}{2}[\boldsymbol{\beta} - \boldsymbol{\beta}_b]^T \boldsymbol{B}_\beta^{-1}[\boldsymbol{\beta} - \boldsymbol{\beta}_b] \\ + \tfrac{1}{2}[\boldsymbol{x} - \boldsymbol{x}_b]^T \boldsymbol{B}^{-1}[\boldsymbol{x} - \boldsymbol{x}_b] \end{aligned} \tag{5}$$

This work uses the temperature and relative humidity profiles obtained from the MWR. Extensive documentation has shown that significant observation errors may arise during the retrieval of relative humidity from ground-based MWR systems [60,61]. Consequently, quality control (QC) of MWR observations becomes imperative for conducting variational data assimilation. In this study, we used the quality control (QC) scheme proposed in [62], which consists of checks for extreme values, time consistency, and vertical consistency. Additionally, before performing data assimilations, we applied a bias correction technique to the MWR profiles using sounding observations:

$$\widetilde{x}_M = x_M + \frac{\sum_{i=1}^{n}(x_M^i - x_{TK}^i)}{n} \tag{6}$$

In Equation (6), $x_M$ denotes the MWR observation before bias correction, $\widetilde{x}_M$ denotes it after bias correction, $x_{TK}$ represents the sounding observation, and n signifies the number of samples. This approach aims to align the MWR observations more closely with the known accuracies of sounding observations, thereby enhancing the reliability of the MWR data used in the assimilation process.

*2.5. Model Configurations and Experiment Design*

The Weather Research and Forecasting model version 4.3 (WRFv4.3) and its associated 3DVAR system were used for the experiment. The simulation covered three nested domains with the center located at (34.47°N, 114.21°E), as depicted in Figure 4. The grid sizes for these domains were 151 × 151, 202 × 151, and 220 × 151, with corresponding grid spacings of 27 km, 9 km, and 3 km, respectively. The model was configured with 50 vertical layers and a top pressure level of 50 hPa. The physical parameterizations selected for this study included: (1) microphysics scheme: WSM6; (2) planetary boundary layer scheme: Yonsei University (YSU) scheme; (3) land-surface model: Noah model; (4) longwave radiance scheme: rapid radiative transfer model (RRTM) scheme; (5) shortwave radiance scheme: Goddard scheme; (6) cumulus scheme: Kain–Fritsch scheme [63] for d01 domain, but with the cumulus parameterization for d02 and d03 switched off. All the simulation experiments share the same sets of parameterizations. Certain schemes, such as the WSM6 microphysical scheme and the Kain–Fritsch cumulus scheme, were chosen based on extensive prior research [64], which indicated their effectiveness in generating heavy rainfall in mid-latitudes. For model simulation initialization, this study employed the NCEP FNL analysis. The observation data were obtained from Chinese surface meteorological stations and archived by the National Meteorological Information Center.

The experiment in this paper started at 12 UTC on 2 July 2022. Data assimilation was conducted after the 6 h spin-up run. FY-4A AGRI and MWR data were assimilated at 1 h intervals during 18–00 on July 2. Finally, a 24 h forecast was performed after each assimilation (see Figure 5 for details). Details regarding four distinct experiments are outlined in Table 2, including an initial control run without data assimilation (CTRL), another incorporating only temperature and humidity profiles from seven MWRs (Test1), a third involving the assimilation of radiance from FY-4A AGRI (Test2), and a final experiment combining both FY-4A AGRI and MWR assimilations (Test3) [29].

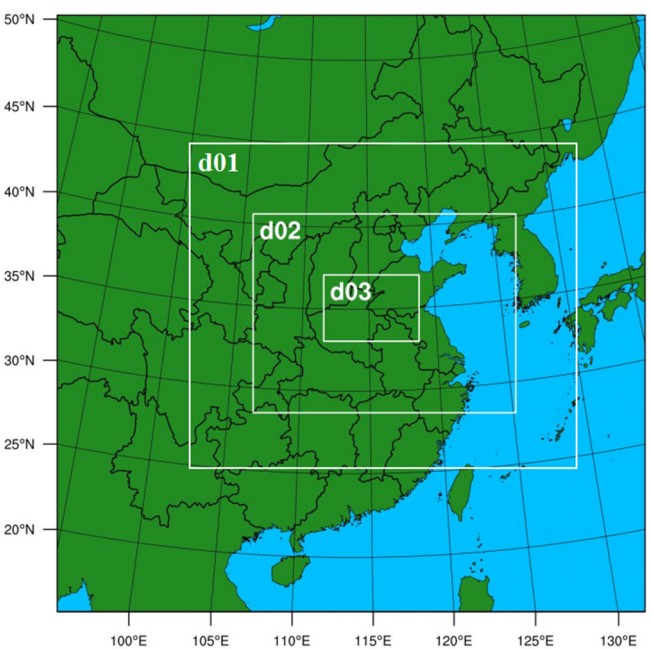

**Figure 4.** Simulation Area of WRF Model.

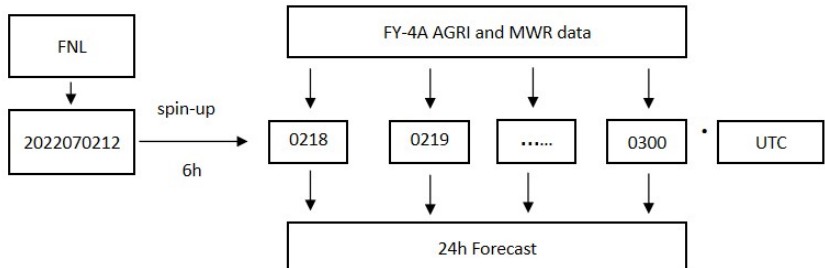

**Figure 5.** Flow chart of the assimilation and forecast of the heavy rainfall case in Kaifeng on 3 July 2022.

**Table 2.** Assimilation scheme.

| Scheme | Assimilated Data | Assimilation Interval |
|--------|------------------|----------------------|
| CTRL | No | |
| Test1 | temperature and humidity profiles from seven MWRs | 1 h |
| Test2 | FY-4A AGRI radiance channels 9–10 | 1 h |
| Test3 | both FY-4A AGRI and MWR data | 1 h |

## 3. Results

### 3.1. The Impact of Data Assimilation on Analysis Field

Data assimilation plays a crucial role in refining the initial conditions for numerical models, thereby enhancing the accuracy of model forecasts. To evaluate the effects of assimilating FY-4A AGRI and MWR data on these initial conditions, an initial condition report at 00 UTC on 3 July 2022 served as the baseline for comparison. The temperature and humidity profiles from a control test (CTRL) were compared against those from three assimilation experiments (Figure 6). According to the results presented in Figure 6, the temperature profile resulting from the combined assimilation of FY-4A AGRI and MWR data demonstrates a closer alignment with observed data across various atmospheric layers (Figure 6a). Specifically, the assimilation leads to adjusted temperatures in the lower atmosphere (below 850 hPa) being warmer and those in the upper atmosphere being cooler, effectively mitigating the warm bias in the lower atmosphere and the cold bias above, thereby rendering them more consistent with observational data.

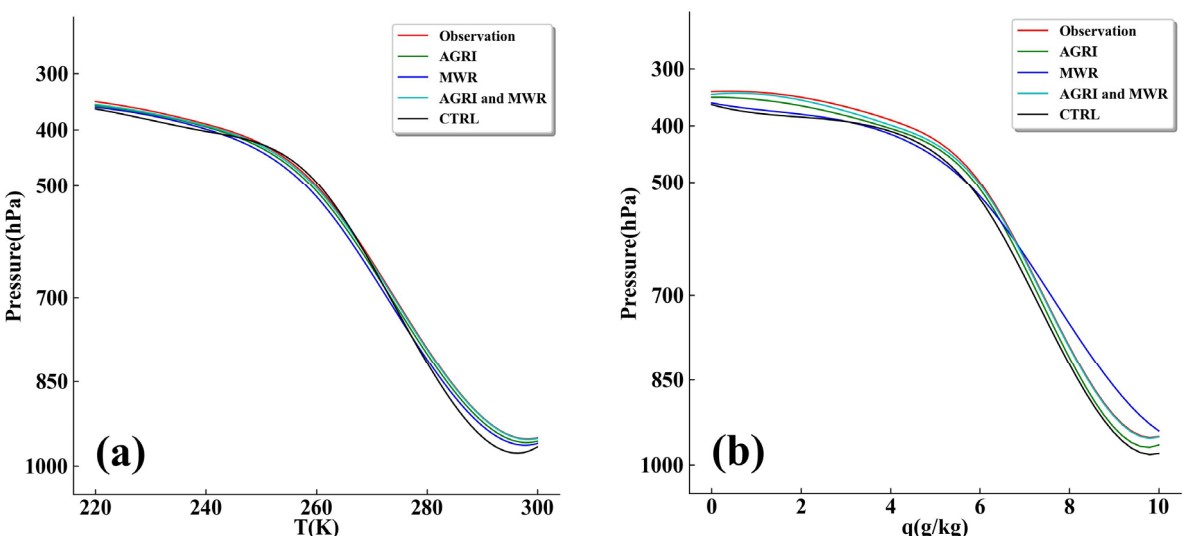

**Figure 6.** Vertical profiles of (**a**) temperature (T; K) and (**b**) water vapor mixing ratio (Qvapor; g·kg$^{-1}$) at 00 UTC, 3 July 2022.

Regarding specific humidity, the atmosphere is observed to be drier above 500 hPa and more humid in the middle to lower layers (Figure 6b). The specific humidity profile obtained from the combined FY-4A AGRI and MWR assimilation experiment addresses these discrepancies, resulting in a profile that more closely mirrors the observed data. While the temperature and humidity profiles from the FY-4A AGRI assimilation alone show a high degree of congruence with observations, an observed increase in Qvapor error below 700 hPa suggests potential inaccuracies in the lower atmospheric layers detected by the microwave radiometer. Overall, the assimilation of FY-4A AGRI and MWR data notably improves the match between the analysis field and the observed field regarding temperature and humidity profiles, underscoring the value of integrating these data sources in enhancing the model's initial conditions.

### 3.2. The Impact of Data Assimilation on Humidity Condition

Figure 7 displays the 500 hPa humidity increments at 14 UTC on 3 July 2022 from the three assimilation experiments. The results reveal that, while the patterns of humidity increments at 500 hPa in the two experiments (Test1 and Test2, shown in Figure 7a,b, respectively) are similar, the specific humidity increments in Test1, particularly over southern Kaifeng, are stronger than those in Test2. In Test2, the humidity increments at 500 hPa are primarily concentrated over areas experiencing heavy rainfall, extending towards the central southern part of Kaifeng. Notably, Test3 (shown in Figure 7c) presents a progressive change in humidity increments, aligning with results from Test2. This indicates that the incorporation of AGRI radiance primarily influences moisture content in the middle-upper atmospheric layers. Furthermore, the assimilation of both FY-4A AGRI and terrestrial MWR data yields beneficial results. This analysis underscores the effectiveness of combining FY-4A AGRI and MWR data in refining initial moisture conditions within model simulations.

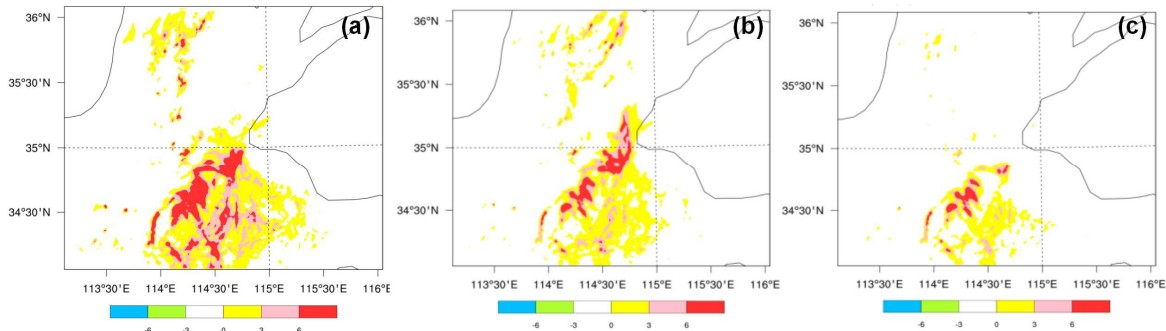

**Figure 7.** The 500 hPa humidity increments (unit: g/kg) at 14 UTC, July 3, 2022, from different experiments: Test1 (**a**), Test2 (**b**), and Test3 (**c**).

### 3.3. The Effects of Data Assimilation on 24 h Accumulated Rainfall Forecast

Evaluating the predictive capability of numerical models is crucial, particularly in forecasting precipitation. Figure 8 presents the 24 h accumulated precipitation results from four different experiments. In the CTRL experiment (Figure 8a), the predicted rainfall in the northern and eastern zones did not reflect the observed intense rainfall (Figure 2), particularly underestimating rainfall in the central region, including Kaifeng. In contrast, the Test1 experiment (Figure 8b), which incorporates MWR data assimilation, adjusted the 24 h rainfall forecast within the Kaifeng region more effectively. However, only minor improvements were observed in the northern and eastern regions compared to CTRL. The Test2 experiment (Figure 8c) aligned more closely with the observed rainfall distribution (Figure 2), correcting the underestimated rainfall near Kaifeng seen in the CTRL experiment. The predicted maximum rainfall center and strength in Test2 closely resemble the actual observations (Figure 2). Furthermore, the Test3 experiment (Figure 8d) enhances both the intensity and distribution of rainfall, particularly in the Kaifeng region, compared to Test2. The 24 h rainfall simulation in Kaifeng closely matches the observed distribution (Figure 2). These findings suggest that the combined assimilation of AGRI radiance and MWR data significantly improves the forecast accuracy of short-term heavy rainfall events.

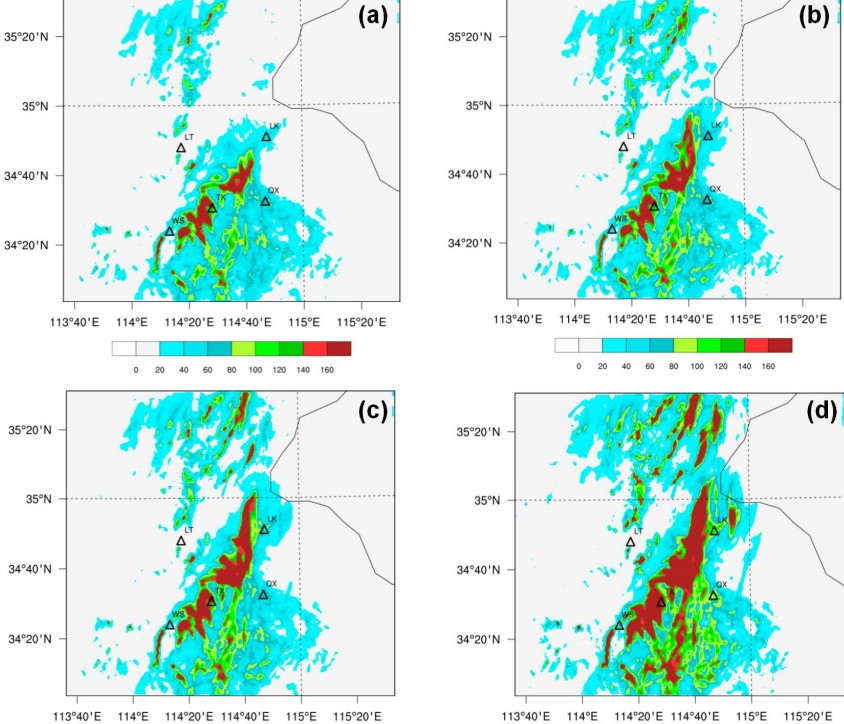

**Figure 8.** Simulated 24 h rainfall accumulation by CTRL (**a**), Test1 (**b**), Test2 (**c**), and Test3 (**d**).

### 3.4. Case Study

Our subsequent analysis focuses on high spatiotemporal resolution data generated by the WRF model after the assimilation of FY-4A AGRI and MWR data. This analysis examines various aspects, including upper-level fields, wind patterns, moisture conditions, mesoscale features, and the influence of topography during the rainfall event.

#### 3.4.1. Height and Wind Fields

Figure 9 illustrates the 200 hPa geopotential height field, wind field, and jet stream region on 3 July 2022. At 02:00, prior to the heavy rainfall (Figure 9a), a broad and deep high-altitude trough was observed over the Mongolian Plateau and southern Gansu Province, exhibiting a north-to-south distribution. A high-altitude jet stream band with core intensity exceeding 50 m/s was located on the southwest side of this trough (35°N–50°N, 90°E–100°E). The Kaifeng area lies southeast of the jet stream region, relatively far from the trough. By 14:00 on July 3 (Figure 9b), as the trough moved eastward and deepened, two jet streams emerged over mainland China. The western jet stream near Gansu reached over 55 m/s, while the eastern jet stream around North China maintained over 45 m/s. Kaifeng is located in the trough's southeastern part, close to the eastern jet stream. The strong, warm, and moist airflow ahead of the trough favors vertical moisture transport. By 20:00 on July 3 (Figure 9c), the trough continued moving eastward and lifting northward, merging the two jet streams into an east–west-oriented band. Kaifeng is in the trough's southwestern part. The northwest sinking airflow behind the trough brings about a transition to sunny conditions. By 08:00 on July 4 (Figure 9d), the main system had moved away from Kaifeng, with a northwesterly downdraft behind the high-altitude trough bringing fine weather.

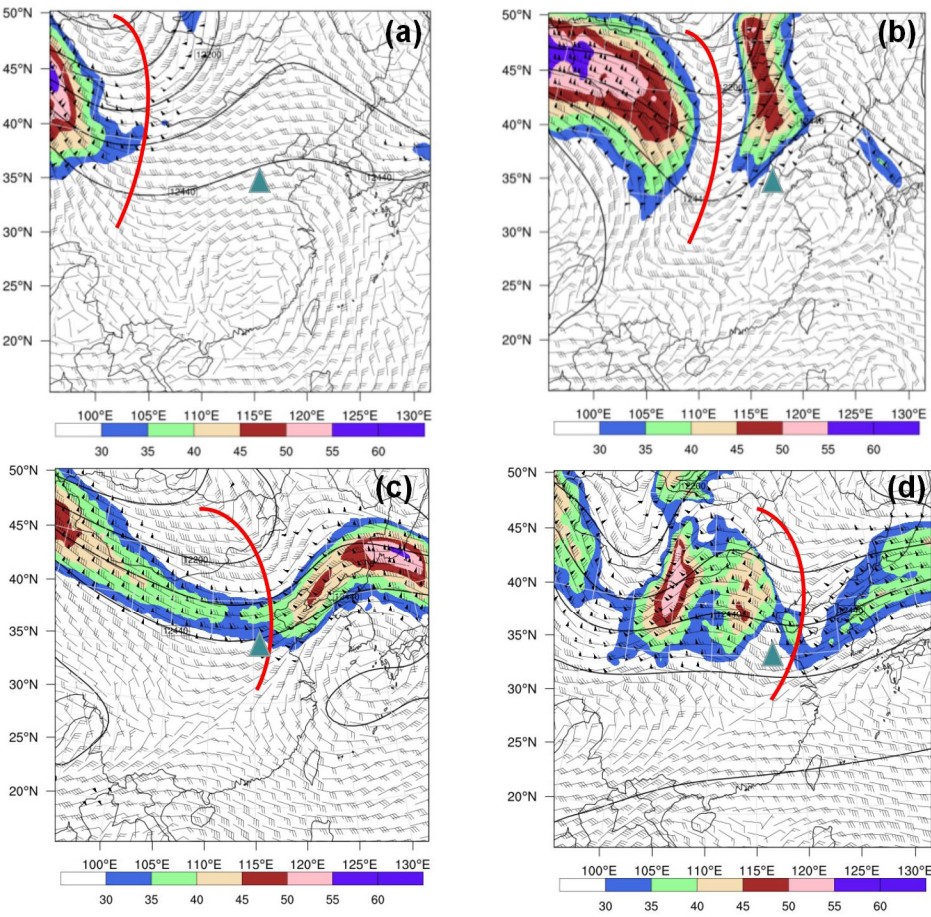

**Figure 9.** A 200 hPa height field (contour line, unit: gpm), wind field (wind plume, unit: m·s$^{-1}$), and jet zone (shaded area in the Figure) at 02:00 (**a**), 14:00 (**b**), and 20:00 on July 3 (**c**), and 08:00 (**d**) on 4 July 2022. The red line represents a high-altitude trough.

### 3.4.2. Water Vapor Condition Analysis

Water vapor conditions play a key role in the formation of rainfall processes. If the humidity is higher and the wet layer is thicker, the more it can promote the formation and development of the rainfall process. Figure 10 shows the vertical profile of relative humidity at 14:00 on 3 July 2022. The profile reveals high humidity (over 90%) between 850 hPa and 300 hPa due to the upward motion of the upper-level trough and the rise of warm, moist air to tropospheric heights. This analysis demonstrates that the Kaifeng area, with its high humidity, provides favorable conditions for the formation and development of heavy rainfall.

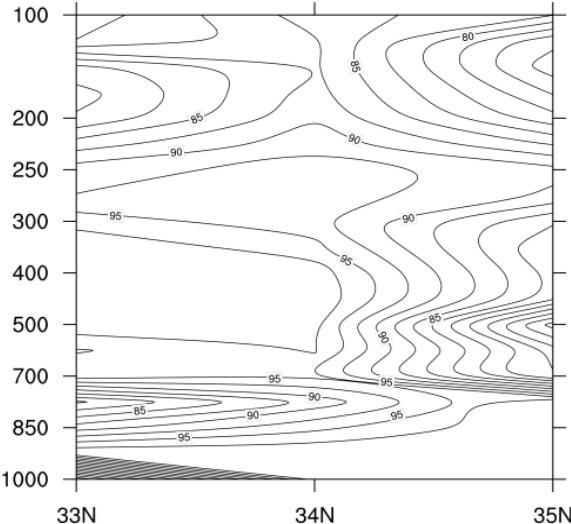

**Figure 10.** Vertical profile of relative humidity at 14:00 on 3 July 2022 (unit: %).

### 3.4.3. Dynamic Condition Analysis

Dynamic conditions are crucial in understanding the development of weather systems, particularly the role of strong upward motion in the vertical direction, which is essential for the formation and development of these systems. This upward motion facilitates the transport of heat, momentum, water vapor, and other factors to the upper atmosphere, leading to precipitation [65]. Figure 11 illustrates the vertical velocity profiles at 02:00 and 14:00 on July 3. The analysis of these profiles indicates significant upward motion around 14:00. Over the Kaifeng area, vertical motion was vigorous from the lower levels to the higher levels of the atmosphere, especially between 500 hPa and 300 hPa, where the vertical motion was strongest. The maximum vertical velocity was observed around 350 hPa, reaching a value of 0.4 m/s. This suggests that during the heavy rainfall period in the Kaifeng area, convective activity is intense. The strong vertical upward motion triggers atmospheric convection, leading to the release of unstable energy in the atmosphere, thereby sustaining and intensifying convective weather.

Figure 12 presents the 350 hPa upper-level divergence field at the same time intervals. At 02:00 on July 3 (Figure 12a), before the precipitation occurrence, the upper-level jet stream above the Kaifeng area was weak, and the divergence motion was mainly concentrated on one side of the jet stream entrance with relatively low intensity. At 14:00 on July 3 (Figure 12b), as the trough deepened and the upper-level jet stream strengthened, the divergence motion in the Kaifeng area intensified. This divergence leads to atmospheric mass adjustment, causing vertical motion below and forcing precipitation. By 20:00 on July 3 (Figure 12c), the maximum divergence center moved to central Inner Mongolia, Hebei, and Henan. As the upper-level jet stream moved northeastward, the divergence region shifted from Henan to Liaoning, leading to a weakening of precipitation over the Kaifeng area (Figure 12d). This provides dynamic conditions for the generation and development of heavy rainfall.

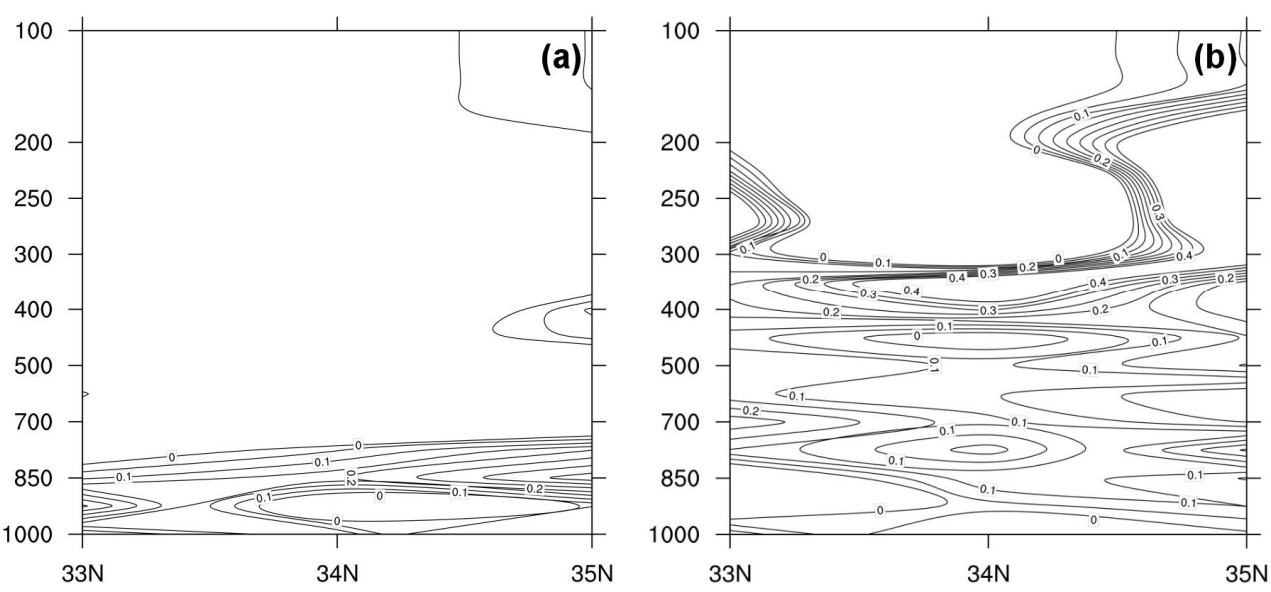

**Figure 11.** Vertical profile of vertical velocity (unit: m·s$^{-1}$) at 02:00 (**a**) and 14:00 (**b**) on 3 July 2022.

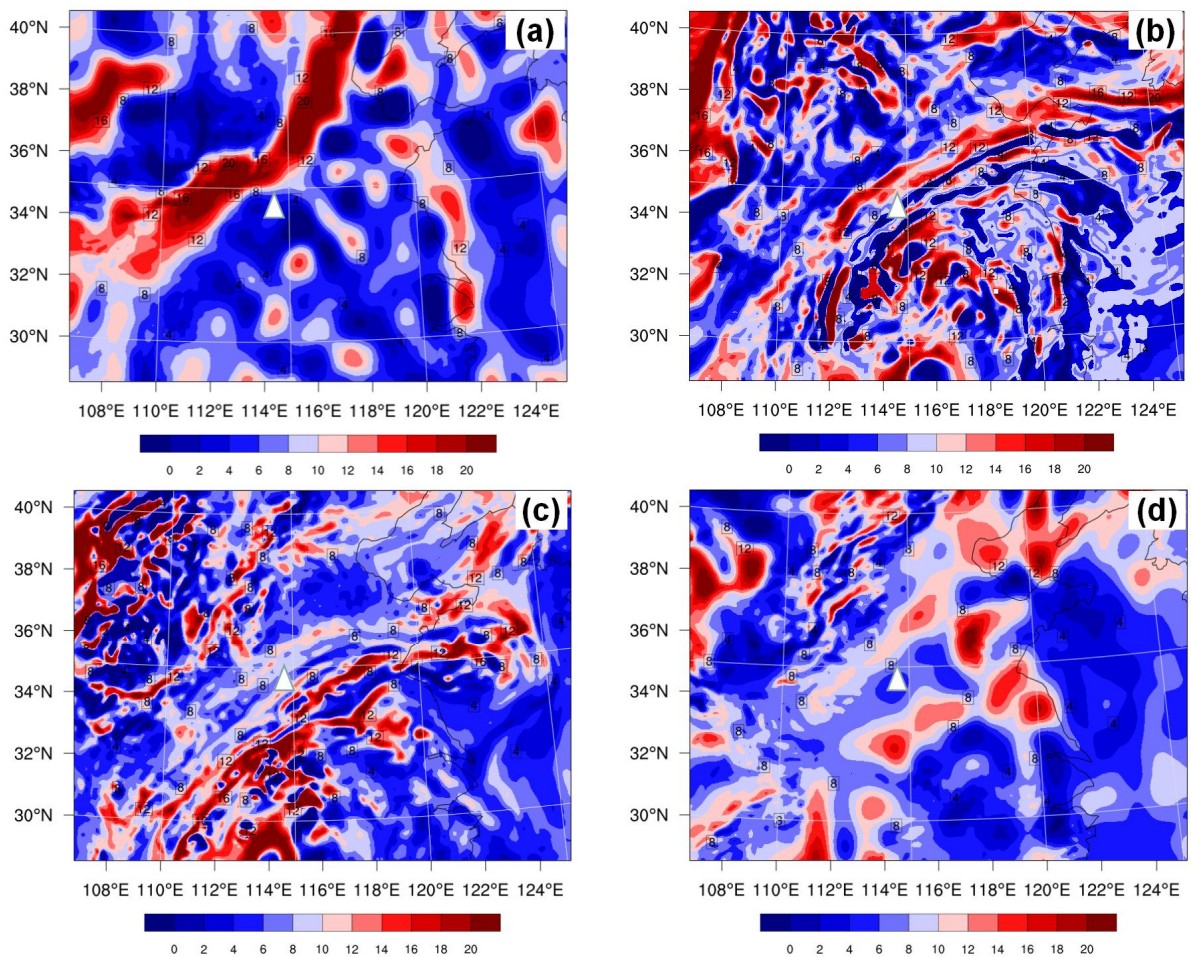

**Figure 12.** A 350 hpa divergence at 02:00 (**a**), 14:00 (**b**), and 20:00 (**c**) on July 3, and 08:00 (**d**) on 4 July 2022 (unit: $10^{-5}$s$^{-1}$).

Equivalent potential temperature ($\theta_{se}$) is a key indicator of atmospheric stability. Figure 13 shows the vertical profile of $\theta_{se}$ at 02:00 and 14:00 on July 3. During the occurrence and development of heavy rainfall, the value of $\partial\theta_{se}/\partial z$ in the Kaifeng area from ground

level to 300 hPa is less than 0, indicating a decreasing trend of $\theta_{se}$ with increasing height. The results illustrate that the middle and lower atmospheric stratification in the Kaifeng area is in an unstable state. There is strong vertical motion, and under the influence of this upward motion, moisture and energy from the lower levels are continuously transported to higher altitudes.

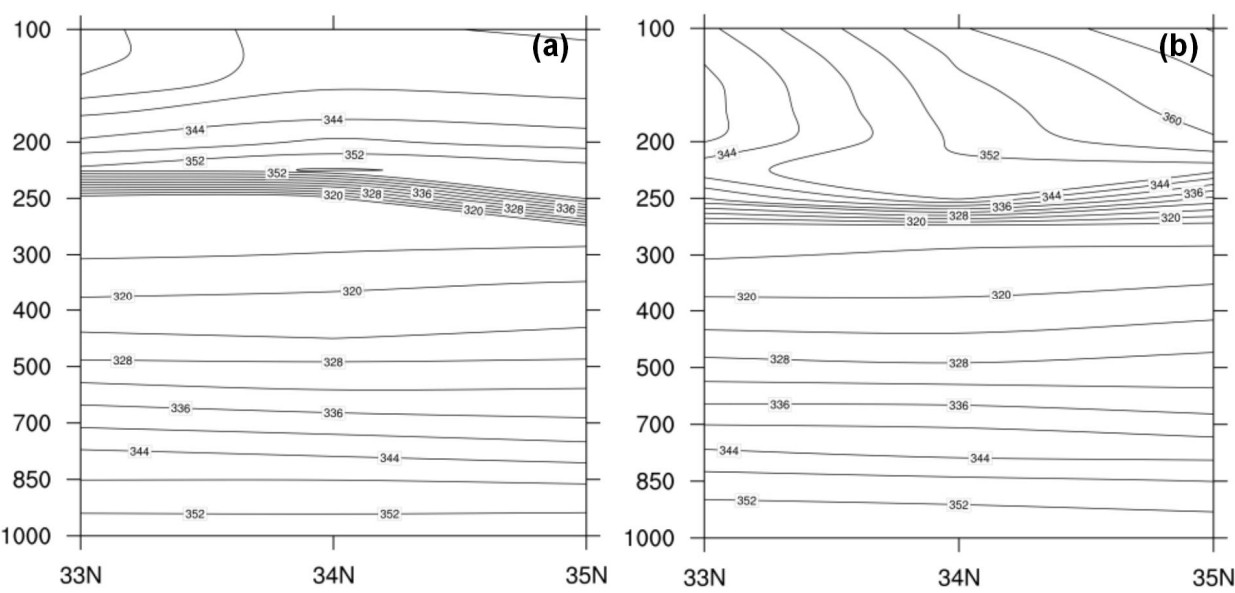

**Figure 13.** Vertical profile of equivalent potential temperature (unit: K) at 02:00 (**a**) and 14:00 (**b**) on 3 July 2022.

Figure 14 illustrates the sounding curves from the Kaifeng station on 3–4 July 2022. At 02:00 on July 3, the relative humidity between 850 hPa and 500 hPa was notably high. The convective available potential energy (CAPE) was measured at 56 J/kg (Figure 14a), suggesting relatively weak convective development and resulting in only light and sporadic precipitation in the Kaifeng area. However, as the upper-level trough moved eastward and intensified, the Kaifeng area was positioned ahead of the trough. Influenced by the warm and moist southwest airflow, humidity in the upper atmosphere over Kaifeng significantly increases, leading to a continuous rise in CAPE. By 14:00 on July 3, CAPE reached 3420 J/kg (Figure 14b), an increase of 3364 J/kg from 02:00. This rapid accumulation and release of unstable energy create favorable conditions for heavy rainfall, coinciding with the precipitation peak in Kaifeng. By 20:00 on July 3, as the trough continued to move northeastward and Kaifeng transitioned to the rear of the trough, CAPE decreased by 1964 J/kg (Figure 14c), marking a shift in precipitation type to convective showers. By 08:00 on July 4, with the trough's exit, CAPE reduced to 0 J/kg (Figure 14d), indicating a transition from an absolutely unstable condition to an absolutely stable state, thus ending the precipitation process in Kaifeng. This sequence demonstrates a consistent relationship between the formation, development, and termination of heavy rainfall and the corresponding phases of CAPE.

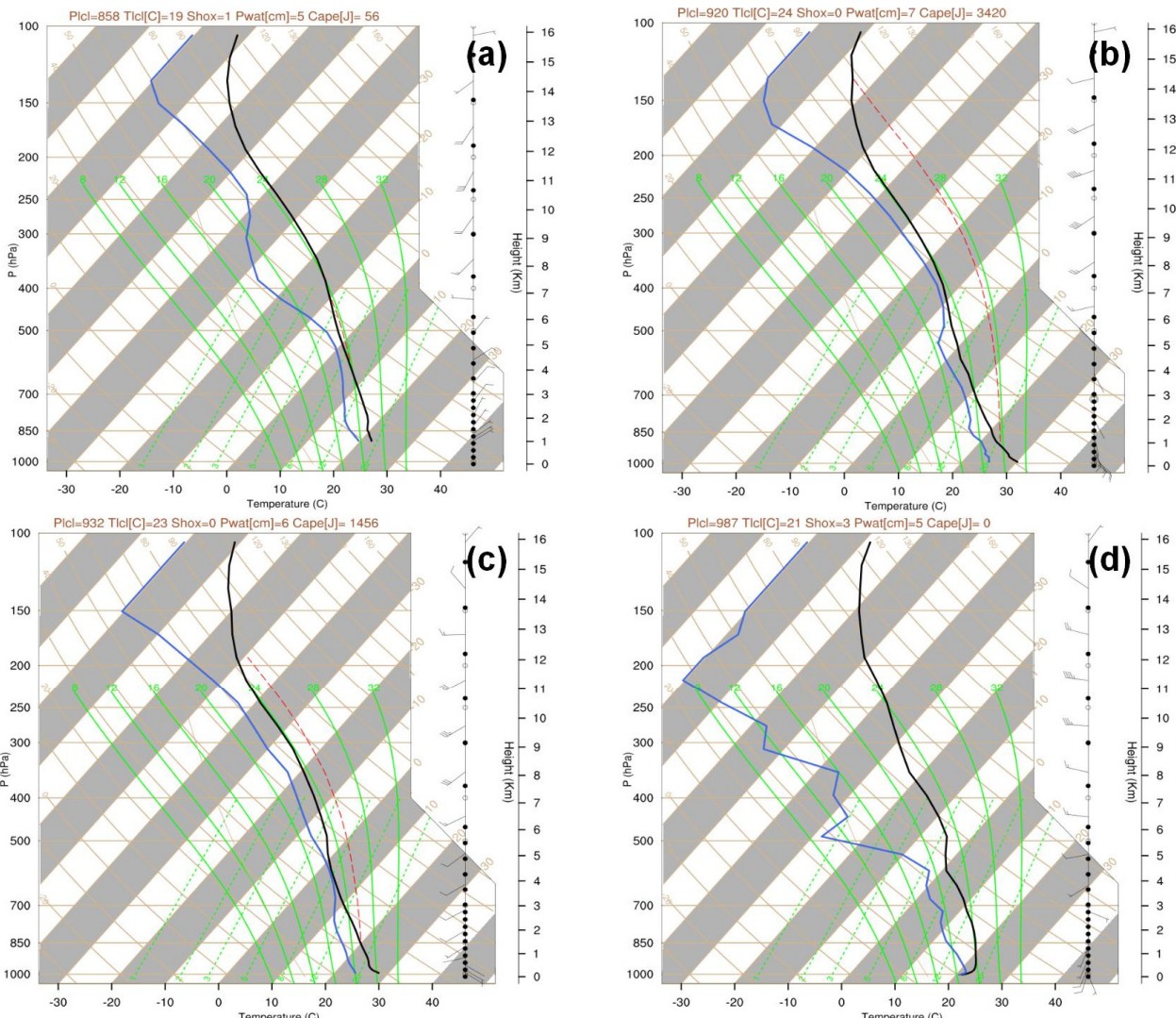

**Figure 14.** Sounding curves of Kaifeng railway station at 02:00 (**a**), 14:00 (**b**), 20:00 (**c**) on July 3, and 08:00 (**d**) on 4 July 2022. Red line: state curve; blue line: dew point temperature; black line: temperature profile.

### 3.4.4. Terrain Condition Analysis

Terrain plays a crucial role in the dynamics of precipitation development. Conducting terrain-sensitive experiments can effectively capture the influence of topographical factors on precipitation processes. Figure 15 shows the topography of Henan Province, revealing a west—high, east—low pattern. The Taihang Mountains, Funiu Mountains, and Dabie Mountains surround the province to the north, west, and south, respectively, with elevations ranging from 1000 to 2200 m. In contrast, the area around Kaifeng in the central and eastern parts of Henan consists primarily of flat plains, mostly below 200 m in elevation. The Taihang Mountains form the eastern boundary of the Loess Plateau and act as a transition zone between the northern mountains and the plains in Henan. These geographical features significantly influence the weather patterns in the region, particularly in terms of precipitation distribution and intensity.

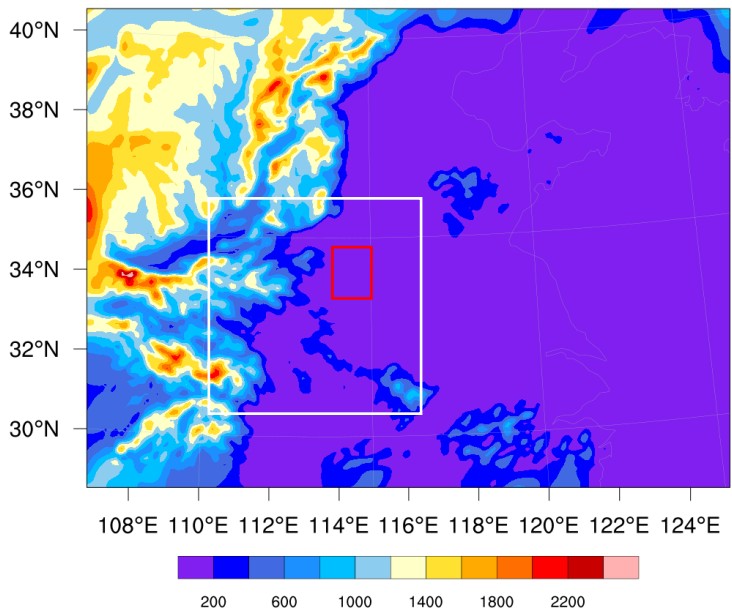

**Figure 15.** Terrain of Henan region (white box represents Henan, and red box represents Kaifeng location).

Experiment Scheme

To investigate the impact of terrain on heavy rainfall, six sets of sensitivity experiments were designed, as outlined in Table 3. All experiments utilize a nested grid scheme with parameterization schemes consistent with the control experiment, named "CTRL".

- ■ Test1: Reduces the elevation of the Taihang Mountains (34.57°N–40.72°N, 110.27°E–114.55°E) by 50%, smoothing the west-to-east elevation gradient.
- ■ Test2: Reduces the elevation of the Taihang Mountains by 75%.
- ■ Test3: Lowers the elevation of the Taihang Mountains by 100%, shifting the transition zone between the plateau and plain to the border area between Shaanxi and Shanxi provinces.
- ■ Test4: Raises the elevation of the Taihang Mountains by 50%.
- ■ Test5: Increases the elevation of the Taihang Mountains by 75%.
- ■ Test6: Elevates the Taihang Mountains by 100%.

**Table 3.** Terrain test scheme.

| Scheme | Changes in Terrain Height/% | Latitude/N | Longitude/E |
|--------|------------------------------|------------|-------------|
| CTRL | 0 | 34.57°N–40.72°N | 110.27°E–114.55°E |
| Test1 | −50 | 34.57°N–40.72°N | 110.27°E–114.55°E |
| Test2 | −75 | 34.57°N–40.72°N | 110.27°E–114.55°E |
| Test3 | −100 | 34.57°N–40.72°N | 110.27°E–114.55°E |
| Test4 | +50 | 34.57°N–40.72°N | 110.27°E–114.55°E |
| Test5 | +75 | 34.57°N–40.72°N | 110.27°E–114.55°E |
| Test6 | +100 | 34.57°N–40.72°N | 110.27°E–114.55°E |

Terrain and Precipitation Analysis

The analysis of 24 h accumulated precipitation from both sensitivity experiments and a control setup underscores the significant influence of the Taihang Mountains' terrain modifications on precipitation patterns in Kaifeng, as illustrated in Figure 16. Decreasing the Taihang Mountains' elevation by 50% from the baseline (control experiment shown in Figure 2) notably reduces the spatial extent of heavy rainfall across Kaifeng. This adjustment eliminates the primary precipitation center previously identified in the northern sector of Kaifeng (Figure 16a), leading to a substantial reduction in precipitation intensity—evidenced by only one out of five national automatic stations recording rainfall

amounts exceeding 140 mm. The predominant precipitation zone shifts to the southern region of Kaifeng.

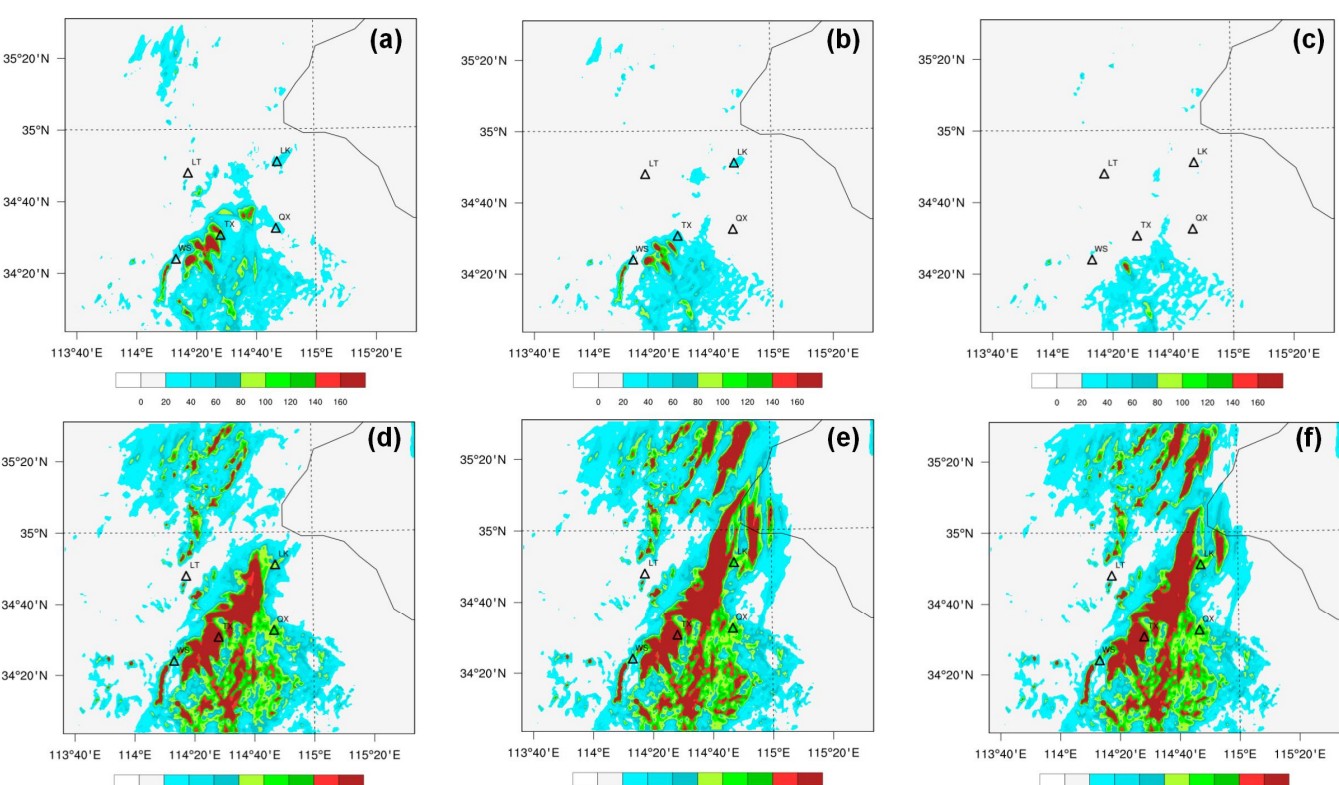

**Figure 16.** Twenty-four-hour accumulated precipitation distribution in Kaifeng area simulated by Test1 (**a**), Test2 (**b**), Test3 (**c**), Test4 (**d**), Test5 (**e**), and Test6 (**f**) from 00:00 on July 3 2022 to 00:00 on 4 July 2022 (unit: mm).

Further elevation reductions in the Taihang Mountains to 75% exacerbate the diminution of rain area coverage and intensity, particularly impacting existing precipitation centers (Figure 16b). A reduction to 100% effectively erases the southern Kaifeng precipitation center, slashing precipitation volumes by 80–100 mm, equating to a total decrease of 50–60% (Figure 16c). Under these conditions, Kaifeng experiences primarily scattered convective precipitation, with heavy rainfall events becoming virtually absent.

Conversely, augmenting the Taihang Mountains' elevation by 50% does not significantly alter the rain belt's orientation, but it amplifies the central precipitation zone, its spatial coverage, and its intensity (Figure 16d). This elevation increase fosters the formation of a new, intense precipitation belt across the middle to northern parts of Kaifeng, marking an overall precipitation increment of 10–20% relative to the control experiment (Figure 2). Elevating the terrain by 75% prompts a notable eastward migration of the intense precipitation belt. The southern Kaifeng precipitation center shifts eastward by approximately 0.5°E, and several rain belts emerge in the northern area, extending in a northeast-southwest orientation (Figure 16e).

Elevation increases to 100% yield precipitation patterns closely mirroring those observed in the Test5 experiment (comparative analysis between Figure 16e,f), suggesting a plateau in precipitation response beyond a certain topographical threshold. At excessively high elevations, water vapor can reach supersaturation due to cooling during ascent, leading to topographic rainfall. Precipitation intensifies with altitude up to a point beyond which diminishing air moisture leads to reduced rainfall amounts, eventually ceasing to increase due to unsaturation at higher altitudes.

This comprehensive analysis demonstrates that the Taihang Mountains significantly impact Kaifeng's precipitation dynamics. Consistently lowering terrain height leads to

diminished precipitation intensity and spatial coverage by 50–60%. On the other hand, terrain elevation increases beyond 50% and enhances the precipitation center, coverage, and intensity by an overall 10–20%. Elevation adjustments exceeding 75% shift the precipitation dynamics eastward by about 0.5°E, without further benefits observed beyond a 100% increase, where precipitation trends stabilize, aligning with the effects seen at the 75% elevation increase threshold.

## 4. Discussion

The current scarcity and uneven distribution of microwave radiometer (MWR) stations in China limit their assimilation impact, leaving the considerable potential of these data largely untapped. However, with the rapid advancements and increasing affordability of ground-based remote sensing technology, it is expected that dense MWR profiling networks will be established, significantly enhancing the accuracy of Numerical Weather Prediction (NWP) models by providing comprehensive atmospheric profiles. The integration of high-resolution Advanced Geostationary Radiation Imager (AGRI) radiance data from geostationary satellites represents a promising avenue for improving short-term forecasts, especially for regional severe convective weather (SCW) events. Our study specifically utilized channels 9 and 10, which are sensitive to the temperature within the 400–600 hPa layer and act as water vapor absorption channels. The minimal discrepancy observed between the brightness temperatures (BT) measured by these channels and those simulated by the model underscores their importance for accurately forecasting severe rainstorms. Consequently, our research focused on assimilating data from these two channels.

Given the indications of a climate transition in China, with potential increases in summer rainfall in northern regions, choosing Kaifeng as the study area is both timely and relevant. This research lays a foundational step towards enhancing SCW warnings and improving disaster prevention and mitigation efforts, showcasing a strategic approach to leveraging AGRI radiance data. While our findings are based on a single heavy precipitation event in Kaifeng, they highlight the utility and potential of assimilating AGRI radiance and MWR data into WRF forecasts. Looking forward, expanding this research to include advanced data assimilation techniques, such as hybrid variational-ensemble methods and data-driven machine learning approaches, could further refine the accuracy of AGRI radiance and MWR data assimilation.

NWP of the meteorological and environmental parameters of important weather events are strongly related to the time and space resolutions, initial conditions, and data assimilation techniques. Over mountainous environments where model grids may not match the height changes because of quickly changing environmental conditions, measurements may include large uncertainties and variability. These uncertainties in the basic thermodynamic parameters can pose a serious challenge when assessing model simulations over mountainous regions. The future work will focus on the extensive measurements of fog, precipitation from weighing and optical gauges, particle spectra from 0.3 lm up to cm size, and visibility, as well as other meteorological measurements such as 3D turbulence and solar radiation collected at the Taihang Mountains and these measurements, representing extreme weather conditions.

It is crucial to acknowledge that the conclusions drawn from this study, while offering valuable insights into the application of regional NWP models, are based on a singular event analysis. This limitation suggests that the results may not universally apply to all weather scenarios. To build a more comprehensive understanding of the impact of assimilating AGRI radiance and MWR data on weather forecasting, future studies should investigate a broader range of cases. Such an expanded analysis will enable a deeper exploration of the benefits and limitations of integrating these data sources into NWP models, potentially leading to significant improvements in forecasting accuracy and reliability.

### 5. Conclusions

This work investigated the effects of integrating AGRI radiance and MWR data assimilation on the accuracy of short-term heavy rainfall forecasts, with a focus on a particular event in Kaifeng. Through a series of data assimilation experiments, this study analyzed the initial conditions and forecasted variables, probing the structural characteristics and mechanisms influencing heavy rainfall from various angles, such as water vapor content, atmospheric dynamics, and topography, among others. The key findings are summarized as follows:

1. Synergistic effect of data assimilation: The joint assimilation of FY-4A AGRI and MWR data yields a notable improvement in forecast accuracy. This synergistic effect corrects the model's warm bias in the lower atmosphere and the cold bias in the upper atmosphere, aligning them more closely with actual observations.

2. Atmospheric conditions favoring heavy rainfall: The analysis reveals that the pre-trough ascent ahead of the upper-level trough induces significant upward movement of warm and moist air, particularly between 850 hPa and 300 hPa, where relative humidity surpasses 90%. This deep moisture layer, along with strong convergence and upward motion, creates optimal conditions for heavy rainfall development.

3. Dynamics of the occurrence and development of heavy rainfall: The study identifies unstable atmospheric stratification in the lower and middle troposphere over Kaifeng during the heavy rainfall event, characterized by strong vertical motion, especially noted between 500 hPa and 300 hPa. A maximum vertical velocity of $0.4 \text{ m·s}^{-1}$ around 350 hPa, coupled with upper-level divergence and lower-level convergence, facilitates vertical ascent. This dynamic setup, together with the accumulation and subsequent release of convective available potential energy, correlates closely with the phases of heavy rainfall development. Moreover, the presence of a strengthening low-level jet at 850 hPa, alongside topographical influences, plays a crucial role in sustaining the rainfall.

4. Topographical impact on rainfall patterns: The terrain of the Taihang Mountains significantly influences precipitation distribution in Kaifeng. Decreasing terrain height leads to a marked reduction in both the intensity and extent of precipitation by 50–60%. Conversely, increasing terrain height by more than 50% enhances the precipitation's center, range, and intensity, with an observed overall increase of 10–20%. Elevating the terrain height by over 75% causes the rain belt to shift eastward by approximately 0.5°E, with a notable eastward movement of the precipitation center. However, further increasing the terrain height beyond 100% does not perpetuate an increase in precipitation; instead, results similar to those observed with a 75% increase are shown.

**Author Contributions:** Y.L. and Z.G. were responsible for conceptualization, supervision, and funding acquisition. J.Z. developed the software and prepared the original draft. J.Z. and Y.J. developed the methodology and carried out the formal analysis. Y.L. and Y.J. validated the data. Z.G., Y.L., J.Z. and Y.J. reviewed and edited the text. J.Z. was responsible for visualization. All authors have read and agreed to the published version of the manuscript.

**Funding:** This study is supported by the National Natural Science Foundation of China (42175082).

**Data Availability Statement:** The NCEP FNL (Final) operational global analysis and forecast data are on 0.25-degree by 0.25-degree grids available at https://rda.ucar.edu/datasets/ds083.3/ (accessed on 6 July 2023). The FY-4A AGRI radiance data and observation data were obtained from ground automatic weather stations operated by the Henan Provincial Meteorological Bureau. The ground-based MWR data can be available from the Meteorological Observation Center of China Meteorological Administration. The Data Assimilation (WRFDA) v4.3 was used to assimilate AGRI infrared radiance and ground-based MWR data, which are available on a server at Nanjing University of Information Science and Technology. The observations are obtained from https://data.cma.cn/ (accessed on 2 July 2023).

**Conflicts of Interest:** The authors declare that they have no conflicts of interest.

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
