# Peer review of "Impacts of Fengyun-4A and Ground-Based Observation Data Assimilation on the Forecast of Kaifeng’s Heavy Rainfall (2022) and Mechanism Analysis of the Event"

_remotesensing, doi:10.3390/rs16101663_

Round 1

Reviewer 1 Report

Comments and Suggestions for Authors

The authors analyze a precipitation event and inspect the value of assimilating the water vapor channel from Fengyun-4A and Microwave ground observations in WRFDA. The topic is relevant and well motivated in the introduction. My concern is that this topic is only partially addressed in the manuscript, in the first part of the results. Then the manuscript shifts to analyze the case study from an observational point of view (not clear what dataset the authors use) and then to conclude, the focus is on sensitivities to elevation representation in WRF. These other aspects are less relevant for this journal wherein the focus is remote sensing. And it seems there is a lack of internal cohesion in the material presented, like gathering together 3 somewhat related topics via the case study. The article can be potentially relevant for remote sensing if the authors focus more in the data assimilation part of the analysis by extending on what they already have to understand better the reasons for the improvements they describe. At a minimum I recommend major revisions, although it appears more appropriate to return it to the authors in order to have more time to complete the research.

Comments on the Quality of English Language

The quality of english language is adequate.

Author Response

Response: We gratefully thank you for your constructive comments and helpful suggestions on our manuscript. The motivation of this study is to employ the WRFDA model with the FY-4A AGRI and MWR data, enhancing the initial and simulated moisture conditions for more precise convective rainfall forecasts of Kaifeng's Heavy Rainfall. At the same time, based on high spatiotemporal resolution data generated by the model after assimilation of FY-4A AGRI and MWR data, this research examines the structural characteristics and influencing mechanisms of heavy rainfall from various perspectives—including water vapor, dynamics, and topography—which not only advances our understanding of extreme precipitation events under complex terrain but also provides a solid foundation for future forecasting and warning efforts. The observation data used in this paper are obtained from Chinese surface meteorological stations and archived by the National Meteorological Information Center. In order to enhance coherence, we have revised the title of the article, which reads: “Impacts of Fengyun-4A and Ground-Based Observation Data Assimilation on the Forecast of Kaifeng's Heavy Rainfall (2022) and Mechanism Analysis of the Event” .

Reviewer 2 Report

Comments and Suggestions for Authors

This paper investigates the impact of data assimilation on numerical weather prediction models. By assimilating AGRI radiation and MWR data, significant improvements have been achieved in the model's humidity profile accuracy, enhancing the forecast of heavy rainfall. A controlled experiment is designed to validate this conclusion. Finally, focusing on a case study of heavy rainfall, the mechanisms influencing the process are analyzed from the perspectives of water vapor, terrain, and other factors.

The content of this paper is comprehensive and well-structured. However, there are several minor issues that need attention:

1. In the abstract section, the author is suggested to introduce the background first, then address the existing problem, and subsequently present the work of this paper. This part requires additional background information.

2. Although the paper demonstrates the improvement of the model by assimilating AGRI radiation and MWR data through controlled experiments, it lacks a mechanistic description. It is recommended to supplement some discussion on why these two types of data were chosen.

3. In the keywords section, "Kai-feng" is inappropriate and is suggested to be removed.

4. In line 580, "workk" should be corrected to "work."

After addressing the above issues, I believe this paper is suitable for publication.

Comments on the Quality of English Language

The author has several spelling errors that need attention.

Reviewer 3 Report

Comments and Suggestions for Authors

The paper presents results from NWP experiments employing data assimilation in Central China. Despite its title only a small part of the paper deals with the improvement as a result of data assimilation. Most of the paper presents only the results of a data assimilated run without comparison with the control and other experiments performed. In this context a change to the paper’s title must be made. Even more the last part (terrain adjustments) presents already well known knowledge.

Despite the inconsistency between the title and the contents of the paper (given that the title will change), the paper itself is well written, easy to follow, with most of the conclusions based on solid scientific arguments.

The paper can be published after minor changes as following:

Lines 1-4: Title change to better represents its contents.

Lines 225-232: Assimilating only clear sky radiance in a case of a major storm might leave only  small percentage of the available original satellite data to finally be assimilated. Can the run be really characterized assimilated?

Lines 311-322 and Figure 6: No information is given on what the “Observation” refers to. Radiosonde? Where?

Figure 12: Since in Figure 11 maximum vertical velocity is located around 350hPa the divergence field should be plotted closer to this level.

Figure 14b: From Figure 1 it can be seen that the storm reached Kaifeng in the morning hours of July 3. At 14:00 (July 3rd) most of the CAPE should have been “consumed”. If this is the case, how can you explain Figure 14b with such high value of CAPE?

Line 580: “workk” à “work”
